# Smoked Tobacco, Air Pollution, and Tuberculosis in Lao PDR: Findings from a National Sample

**DOI:** 10.3390/ijerph16173059

**Published:** 2019-08-23

**Authors:** Anne Berit Petersen, Natassia Muffley, Khamphithoun Somsamouth, Pramil N. Singh

**Affiliations:** 1Department of Graduate Nursing, School of Nursing, Loma Linda University, 11262 Campus Street, Loma Linda, CA 92350, USA; 2Center for Health Research, Loma Linda University, 11234 Anderson St, Loma Linda, CA 92354, USA; 3Center for Information and Education for Health, Ministry of Health, Simuang Road, Vientiane, Lao PDR; 4Transdisciplinary Tobacco Research Program, Loma Linda University Cancer Center, 11234 Anderson St, Loma Linda, CA 92354, USA

**Keywords:** indoor air pollution, cooking fires, biomass fuels, crop, tuberculosis, tobacco

## Abstract

In 2017, more than half of the global burden of incident tuberculosis (TB) came from the Western Pacific region. In Lao People’s Democratic Republic (PDR), the high rates of tobacco use and use of polluting biomass fuels for cooking (e.g., wood, charcoal, crop waste, dung) represent significant risk factors for TB. The purpose of this study was to determine the association between self-reported (1) smoking and TB; and (2) exposure to air pollution (from both cooking fires and environmental tobacco smoke) and TB among adults in Lao PDR. We analyzed data from the 2012 National Adult Tobacco Survey (NATSL) of Lao PDR—a multi-stage stratified cluster sample of 9706 subjects from 2822 households located in all 17 provinces. Utilizing a nationally representative sample and inferential, multivariable methods, we observed a significant increase in odds of self-reported TB among those who smoked tobacco (OR = 1.73, 95% CI = (1.00 to 2.98)). Larger multivariable models identified independent contributions from exposure to tobacco pipes (OR = 21.51, 95% CI = (6.34 to 72.89)) and communal outdoor fires (OR = 2.27, 95% CI = (1.15 to 4.49)). An index measuring combined exposure to smoked tobacco, environmental tobacco smoke in enclosed workspace, indoor cooking fire, trash fires, and other outdoor communal fires also showed a positive association (OR per added exposure = 1.47, 95% CI = (1.14 to 1.89)). The findings of this study underscore the need for multi-sectoral collaboration between tobacco control, environmental health, TB prevention and treatment programs, national authorities, policy makers, civil groups, and the private sector to address the convergence of potential risk factors impacting respiratory health in Lao PDR.

## 1. Background

Although curable and preventable, tuberculosis (TB) is now the leading cause of death from an infectious agent worldwide, surpassing HIV/AIDS [1]. The World Health Organization (WHO) estimates that in 2017, there were approximately 10 million new cases and 1.3 million deaths from TB [1]. A quarter of the world’s population has latent TB infection; however, the majority of active TB cases and deaths continue to occur in low- and middle-income countries. In 2017, 62% of all new cases globally occurred in the Western Pacific region [2].

Complex pathways of inter-related biomedical, social, behavioral, environmental, and structural risk factors for TB have been identified [3,4,5]. At the population level, poor living and working conditions, including poor ventilation, overcrowding, and exposure to indoor air pollution continue to be significant risk factors for TB transmission and progression [5,6]. Other major risk factors include those that impair one’s defense against TB infection and disease, such as HIV infection, malnutrition, diabetes, and tobacco and alcohol use [1,4,6,7]. Indoor air pollution and tobacco use are two risk factors that play significant roles in TB risk profiles at both the individual and population level.

Tobacco use is an established, modifiable risk factor for both latent TB infection and TB disease [7,8,9]. A meta-analysis of 24 studies on the effects of smoking on TB, by Bates et al. (2007), showed that the relative risk of TB disease (RR = 2.3–2.7) was higher among smokers when compared to nonsmokers [8]. Additionally, there is clear evidence that smoking is associated with increased risk of latent TB infection and death with active TB when compared with never-smokers [4,7,8]. Exposure to environmental tobacco smoke (ETS) has also been linked to an increased risk of latent TB infection [9,10,11,12]. While children continue to experience the greatest risk, a recent systematic review of 18 observational studies conducted among adults and children found an increased relative risk of latent TB infection and active TB with exposure to ETS, after controlling for age, use of biomass fuel in the home and household contact with active TB [11]. Finally, there is increasingly strong evidence for an association and a large potential impact for a causal relationship between indoor air pollution and TB infection among both adults and children [12,13].

For the present study, we focused on the Lao People’s Democratic Republic (Lao PDR)—a landlocked country in the Western Pacific region that has a high burden of TB infection, smoking, indoor air pollution, and outdoor air pollution [1,14]. In 2012, the estimated prevalence of TB (all cases) was 514 per 100,000 people, and there was a 32% case detection rate of TB among the population [14,15]. For Lao PDR we have nationally representative data on smoking, air pollution, and self-reported TB sampled from 9043 adults from the 2012 National Adult Tobacco Survey of Laos (2012 NATSL) that was designed as part of the US NIH/Fogarty Asia Tobacco Control Leadership program [16,17,18]. This survey included a special supplement on indoor and outdoor air pollution supported by pilot studies conducted by the same investigators [19,20]. Per prior reports of the NATSL, the self-reported prevalence of smoking among individuals was 43.1% for men and 8.4% for women [16] and self-reported rates of exposure to individual sources of air pollution which are concordant with past estimates were as follows [21]: 74.5% ETS exposure at home, 78% exposure to indoor cooking fires, 56% exposure to outdoor cooking fires, 38.6% exposure to trash fires, and 30.1% exposure to crop fires. [22].

Our general aim is relate these highly prevalent respiratory disease risk factors to TB in the NATSL sample. Our specific aims are (1) to determine whether smoked tobacco contributes to higher rates of TB; (2) to determine whether indoor air pollution sources (cooking fires, ETS) contribute to higher rates of TB; (3) to determine whether outdoor air pollution sources (crop burning, trash fires) contribute to higher rates of TB; and (4) to determine whether the combination of smoking, indoor air pollution, and outdoor air pollution incrementally contribute to higher rates of TB.

## 2. Methods

### 2.1. Study Population

The 2012 NATSL enrolled 9721 subjects ages 15 years and older who were present at home and consented to survey participation [16]. Using the 2008 census as a sampling frame, the 2012 NATSL sample was selected using a stratified, multi-stage cluster sample [17]. Lao PDR was stratified into 17 census-derived survey domains which represented the 17 provinces of the country. For the first stage of sampling, primary sampling units from each domain (i.e., villages or comparable urban unit) were selected to ensure 90% statistical power to estimate national prevalence within 2% accuracy. Each primary sampling unit was converted into number of households and number of enumeration areas. An enumeration area was selected from each primary sampling unit based on the probability proportional to the enumeration area size. A list of households in each selected enumeration area served as the sampling frame for selection of households. Enumeration areas that had more than 200 households were segmented, and only one segment was randomly selected for inclusion in the survey.

Survey teams consisted of four or five trained interviewers and enumerators who worked in each of the 17 census derived regions described above. For specific procedures of administering 2012 NATSL survey items and pictograms, a total of 86 people and two of the report authors were trained during a one-week session that preceded the data collection efforts. The sampling method consisted of 9721 adults selected from households inclusive of all private and single member households from all provinces. The survey did not include institutional households such as military barracks, prisons, hospitals, and residents of temples. Written informed consent was obtained from each subject, and the protocols for the 2012 NATSL national survey and its sub-studies were approved by the Institutional Review Board of Loma Linda University (IRB #5170182). An incentive of roughly US $0.50 was offered to each participant. The final sample size of the study was reduced to 9043 people due to missing data for self-reported TB.

### 2.2. Questionnaire

The questionnaire for the 2012 NATSL was designed using (1) qualitative studies to determine items on tobacco use and other lifestyle variables and to obtain representative pictures for pictograms [23]; (2) standardized items from the Global Adult Tobacco Survey (GATS) of the Global Tobacco Surveillance System [24]; and (3) the 2006 national survey of tobacco use in Lao PDR [25]. The final survey contained sections on demographics, smoked tobacco, secondhand smoke exposure, air pollution exposure, and current health status, including TB diagnosis. Pictures and pictograms were used to measure various variables such as the type of tobacco smoked, size of handrolled cigarettes, type of waterpipe smoked and where water was obtained, exposure to indoor and outdoor cooking fires, trash burning, communal burning of crops, and indoor tobacco smoke. A local cartoon artist was hired to produce thepictograms for the questionnaire (Figure A1 and Figure A2). The pictures and pictograms were used to help participants understand the exact nature of the question to ensure accuracy. Tuberculosis was measured in the survey with the question, “Has a doctor or other health worker EVER diagnosed or told you that you are suffering from the following: (a) Tuberculosis.”

The content and design of the survey was completed in English by an international team of investigators. The final survey was conducted in the local language, and the written survey items were translated and back-translated to verify content, criteria and semantic equivalence by bilingual and monolingual experts who used the methods described by Flaherty et al. [26]. During data collection, manual editing was performed by field supervisors from the National Institute of Statistics. After the survey was returned, the data entry, verification, coding, and data cleaning was accomplished using the Census and Survey Processing System (CSPro, Suitland, MD, USA) software package.

### 2.3. Statistical Analysis

The smoked tobacco use exposure variables used in the analysis were created using a standardized coding method for the Global Adult Tobacco Survey (GATS) items that classifies subjects as daily smokers, less than daily smokers, and non-smokers—both currently and in the past. Prevalence of exposure to indoor and outdoor air pollution and smoked tobacco in the national sample were reported for pertinent demographic variables, which included: age, gender, village type, income, and education. Age-adjusted odds were calculated for both smoked tobacco and various sources of air pollution. Inferential, multivariable methods using logistic regression tests of exposure and trend were used to test for effect. A multivariable logistic regression model, in which smoked tobacco and self-reported exposures to air pollution sources were the exposure variables, and self-reported TB was the outcome, was also used to investigate the relationship between TB and the combined effect of tobacco smoke and air pollution. Gender, village type, income, and education were also added to the multivariable model to control for confounders. For checking the assumptions of the logistic regression model we note that the case rate of TB in the analysis allowed for up to 15 variables in a multivariable model and this was not exceeded in our analyses. Model fit of continuous and indicator terms in the logistic model was checked through the Lemeshow–Hosmer Statistic and log likelihood ratio tests, respectively.

To account for the stratified, multi-stage cluster design, the variances for calculating 95% confidence intervals for measures of prevalence were computed using a Taylor series linearized method. This method allowed for the computation of between-cluster variance estimators that accounted for the intra-cluster correlation among subjects within the same village. Point estimates were further adjusted by sample weights to account for different sampling fractions within each of the 17 domains. Statistical analyses were performed with SURVEYFREQ and SURVEYLOGISTIC modules of SAS 9.4 (SAS Institute, Raleigh, NC, USA).

## 3. Results

### 3.1. Demographics and Tuberculosis

In Table 1, we provide the demographic characteristics of all subjects from the 2012 National Adult Tobacco Survey of Lao PDR (2012 NATSL) who had complete data on self-reported lifetime history of tuberculosis (*n* = 9043). Most subjects were young and middle aged adults (74.35% ages 15–44 years), rural dwelling (68.25%), and had completed 12 years of education or less (93.8%). A little over half of all subjects were from households earning 3 USD per day or less. In the total 2012 NATSL sample, we found a prevalence of ever having been diagnosed with TB of 1.67% (95% CI, 1.44 to 1.90). Using the census derived sampling weights, we computed that this weighted prevalence estimate represented 52,864 cases (95% CI, 45,506 to 60,223) of lifetime TB infection among adults ages 15 years and older during 2011.

### 3.2. Association between Demographic Variables and Lifetime History of Tuberculosis in 2012 NATSL

We found a significant positive association (0.3% increase in odds per year) between self-reported lifetime history of TB and age (OR per year = 1.003, 95% CI = (1.001 to 1.005)). In Table 2, we provide associations with lifetime history of TB for demographic variables indicating that female gender, higher income, and higher education trend toward protection against TB.

### 3.3. Association between Exposure to Smoke and Tobacco in 2012 NATSL

In Table 3, we provide univariable associations with self-reported lifetime history of TB for smoked tobacco and various sources of air pollution. The major findings indicate a significant 73% increase in odds of TB for all forms of smoked tobacco, and that this was especially evident for tobacco pipes (OR = 21.03, 95% CI = (5.53 to 79.88)), and hand rolled cigarettes (OR = 1.68, 95% CI = (0.90 to 3.11)).

Among the reported exposures to household sources of air pollution, indoor cooking fires (OR = 1.61) and trash burning (OR = 1.69) trended toward a non-significant positive association with lifetime history of TB. ETS exposure in the home was also tested in non- and never-smokers and did not show evidence of association. Among the reported non-household sources of exposure, communal/occupational exposures, outdoor communal fires (OR = 2.31) and ETS exposure in enclosed workspaces (OR = 1.93) were found to have significant and marginally significant associations, respectively.

In a larger multivariable model that adjusted for age and gender, tobacco pipes (OR = 21.51 95% CI = (6.35 to 72.89)) and communal outdoor fires (OR = 2.27, 95% CI = (1.15 to 4.49)) retained significant associations. We then tested an overall index that summed self-reported exposure across the five major effects in Table 3 (smoked tobacco, ETS in enclosed workspace, indoor cooking fire, trash fires, other outdoor communal fires) and found an OR per added exposure of 1.47, 95% CI = (1.14 to 1.89) in a multivariable model that adjusted for age, gender, education, and income.

## 4. Discussion

Utilizing a nationally representative sample of adults in Lao PDR, we identified a statistically significant increase in odds of self-reported TB with use of all forms of smoked tobacco and a very strong effect with tobacco pipes. Self-reported exposure to multiple sources of indoor (cooking fires, ETS in enclosed spaces at the workplace) and outdoor (communal fires, trash burning) air pollution each trended toward about a two-fold higher rate of lifetime TB. When we considered the combined effect of smoking + ETS in enclosed workspace, + indoor cooking fire + trash fires + other outdoor communal fires we found a significant 47% increase in odds of TB for each additional exposure reported (up to five).

When considered collectively, our findings link highly prevalent individual and community based risk factors for respiratory health with the national TB burden in Lao PDR. It is noteworthy that our estimate does not include the contribution to this effect from fine particulate air pollution that comes from coal-burning, as well as vehicle and industrial emissions which are emerging in Lao PDR and have long been part of cross-border effects from neighboring industrialized nations [27,28].

### 4.1. Tobacco and TB

In this study, all forms of smoked tobacco were associated with a significant increase in odds of TB. According to the literature, both former and current smoking increases the risk of acquiring a TB infection, progression to active TB, recurrence of the disease, development of more severe forms of TB, and dying from TB. Also, smoking can impair response to treatment and interfere with adherence to TB treatment, particularly among elderly [7,29,30]. At the same time, quitting smoking has been associated with a significant decreases in TB-related mortality. A study conducted among individuals with HIV found a 65% decrease in TB-related mortality among those that quit compared to those that continued to smoke [31]. These findings underscore the importance of integrating smoking cessation services into TB treatment programs.

The World Health Organization [32] and the International Union Against Tuberculosis and Lung Disease provide guidelines [33] for smoking cessation treatment within TB programs. A 2018 systematic review found that interventions provided during TB treatment are feasible and effective in reducing smoking rates among patients [34]. However, considerable variation in treatment components, provider engagement, and time requirements across programs was observed revealing a critical need for further empirical and qualitative research to inform effective integration of cost-effective interventions into standard practice, particularly in low-resource settings. Additionally, there is a need to provide structural support and training to health care providers on the effect of smoking on TB treatment outcomes and approaches to smoking cessation for patients with TB, as studies indicate that significant knowledge gaps persist [35,36].

### 4.2. Air Pollution and TB

The findings from this national sample indicate that multiple respiratory stressors may exist at a very high prevalence in Lao PDR, particularly in rural settings. More than half of the sample reported daily exposure to ETS and very high rates of exposure to smoke from biomass fuels (indoors and outdoors) all of which have the potential to increase exposure to fine particulate matter, commonly referred to as PM_2.5_ exposure, which in turn weakens the respiratory epithelium and increases risk for respiratory diseases, including TB [4,37,38]. Additionally, although not statistically significant, it is also noted in this analysis that the association between ETS exposure was present among those who had never smoked as well as in active smokers, suggesting that the risk is likely to be compounded for those that smoke. These results support findings from a pilot study conducted by Lopez et al. (2014) in Lao PDR among rural men which found a high prevalence of self-reported chronic exposure to multiple sources of fine particle air pollution (animal handling, dust/dirt, smoke) and a high prevalence of impaired lung function [19]. The large effect observed in this national study continues to raise concerns about the potential compounding threat that exposure to multiple indoor and outdoor sources of air pollution represents to respiratory health in this setting.

At the country level, exposure to fine particulate matter pollution (from indoor and outdoor sources) continues to be strongly inversely related to a country’s level of social and economic development [27]. Additionally, poor living conditions and poverty remain primary drivers behind high levels of PM_2.5_ exposure [39]. The burning of biomass fuels (e.g., wood, crop residue, dung, and charcoal) continues to be a leading source of fine particulate pollution indoors, which also contributes to outdoor (ambient) air pollution [27]. While the use of biomass fuels is declining globally, in 2017, approximately 3.6 billion people continued to be reliant on these fuels in open fires as their main source of fuel for heating and cooking [27,39]. The lack of access to clean fuels and technologies is due primarily to poverty, and a lack of infrastructure and resources required to deliver and maintain alternative energy sources [27,38]. Even in China, where a rapid transition in residential energy from solid fuels to electricity has occurred in recent decades, the transition tends to lag in rural areas, and transitions are incomplete, with “stacked energy” use being normative (i.e., mixed use of multiple energy) [40]. In 2012, despite introductions of cleaner fuels, 85% of households in China used some form of biomass fuel for heating [41].

Household air pollution, which includes both exposure to smoke from cooking fire and tobacco is now the leading cause of disability-adjusted life years (DALYs) in Southeast Asia and the Western Pacific region [42]. Furthermore, as has been demonstrated elsewhere, in Lao PDR the adverse health effects caused by indoor air pollution tend to manifest more in women and children and women are more likely to be exposed to these pollutants than men because they spend more time indoors and in the kitchen [21,38,43]. For those that continue to rely on polluting fuels, there is a clear need for clean household energy solutions to protect public health. Three of the indicators of the United Nation’s Sustainable Development Goals are closely related to air pollution, health, and access to affordable, reliable, and modern energy sources [44]. The WHO is currently developing a Clean Household Energy Solutions Toolkit (CHEST) to promote clean and safe interventions in the home. The goal of this project is to provide a dynamic analytical framework by which health care professionals, policy-makers and industry stakeholders can design policies and programs that deliver “genuine and substantial health gains” [45]. It is understood that solutions will need to be based on country-specific data on actual health risks, current household energy use, and stakeholder analyses.

### 4.3. Policy-Relevant Implications for Respiratory Health Programs in Lao PDR and the Western Pacific Region

In the Western Pacific region, which already has the highest rate of smoking in the world, we find that rural regions of Lao PDR also have a very high prevalence of indoor and outdoor air pollution. What is not measured is the industrial air pollution from neighboring nations and emerging industrialization in Lao PDR, which, based on the literature, only threatens to compound the current risk profile [46]. These converging factors speak to the need for comprehensive respiratory health policies in Lao PDR that address both smoking and exposures to varied sources of air pollution, particularly for rural settings.

Ending the TB epidemic is one of the health targets of the WHO Sustainable Development Goals, and in 2014, the WHO endorsed The End TB Strategy which aims to ‘end the global TB epidemic’ by 2035 [47]. According to recent mathematical modeling, combating exposure to tobacco smoke is crucial to reducing the risk of TB at the population level, particularly considering the projected increase in smoking rates across many low- and middle-income countries [7,48]. Therefore, to meet these targets, there is a critical need for policy makers and development agencies to advocate for collaboration between sectors addressing indoor air pollution, tobacco control, and TB initiatives. The key principles of the WHO strategy for success include developing strong coalitions between civil society organizations and communities [2].

During the past several decades, TB interventions have tended to focus primarily on increasing delivery and access to treatment services and improving biomedical solutions; however, recently there have been increasing calls to shift from a focus on biomedical models to biosocial models to curb the TB epidemic [5]. Social factors continue to be drivers of the transmission and progression of the disease. TB has long been viewed as a disease of poverty. Not only is TB caused by conditions inherently associated with poverty, including malnutrition, overcrowding, and indoor air pollution, it also leads to poverty due to the financial burden associated with lost wages and cost of treatment. Ortblad et al. (2015) describe these linkages as the a “tuberculosis-poverty cycle” that cannot be broken by biomedical models alone, but rather requires social, environmental, and economic interventions that address associations between development (sanitation, improved water, access to electricity, urbanicity, malnutrition, and education), poverty, health system access, and TB [5,49]. National TB Programs have attempted to implement some schemes for expanding social protection, including compensating the financial burden associated with illness; however, these have tended to be project-based and rely on external funding. There is a need for scalable integrated programs that link national authorities (e.g., Ministries of Energy, Environment, Finance, Welfare, Labor, and Health) with one another and communities [50].

The findings from this nationally representative study underscore the need for population level multi-sectorial approaches to improving the respiratory health in Lao PDR. Based on the literature, both the population’s reliance on biomass fuel, particularly in rural settings and the high rates of smoking (>43% among males according the NATSL) are currently contributing to the alarming rates of TB transmission and progression. Studies have demonstrated that interventions on smoking and indoor air pollution (proximal factors) can accelerate TB decline [51]. There is an opportunity for community development to join forces with tobacco advocacy and TB initiatives to collectively address living conditions, risk behaviors, and social factors common to each of these concerns [3,5]. Furthermore, there is a need to engage the agricultural ministry and the non-governmental organization sector to reexamine current farming and food production practices that represent a threat to respiratory health. As other authors have advocated, the time has come to view eradicating TB as a development imperative which prioritizes poverty alleviation and promotion of sustainable development along with enhanced biomedical approaches [5,50].

As findings in this study suggest, there may be additional interaction effects between smoked tobacco and community sources. Assessing the combined and interaction effects will require more complex statistical models, better exposure measures, and larger samples. There is also a need for in depth environmental studies with sensitive measures of air pollution. These should include a direct measure of personal exposure to particulate matter with diameter of ≥2.5 µm (PM_2.5_) in and around the home. Finally, the literature on mathematical modeling of social and environmental determinants of TB is limited, however, due to the complexity of pathways involved this will require multidisciplinary collaboration between social scientists, epidemiologists, economists, policy makers, and communities affected by TB [3].

### 4.4. Limitations

One of the limitations of the study was that we did not have sufficient statistical power to completely assess interactions between smoked tobacco, indoor air pollution, and all sources of outdoor air pollution. The use of binary measurements may have limited the ability to detect relationships between variables, led to underestimation of variation in groups, and concealed any non-linearity between variables and the outcomes [52]. Thus, some of our findings that combined factors into a model may be subject to type 2 error. The history of TB relied on self-report and did not allow for distinction between latent TB infection versus active TB disease. Additionally, exposure to indoor air pollution and other sources of smoke exposure within the household and in the community also relied on self-report. Furthermore, with the observed association between ETS exposure and self-reported TB, we are unable to determine whether the increased reports of TB are associated with increased contact with individuals with active TB versus ETS exposure. Future studies would be strengthened by use of contact network models and the use of biomarker data.

## 5. Conclusions

Among adults in Lao PDR, we found significant increases in odds of TB for smoked tobacco but non-significant increases in odds of TB for various sources of air pollution. The combined effects of both tobacco smoke and air pollution also indicated increased odds of TB. Our data from a national sample further links these specific domains of high prevalence exposures with TB. The findings call for a multi-sectoral systems approach of bringing tobacco, environmental, health, and TB sectors together to address these converging threats to public health in Lao PDR.

## Figures and Tables

**Table 1 ijerph-16-03059-t001:** Demographic characteristics of the 9043 adults (ages 15 and older) of the 2012 National Adult Tobacco Survey of Lao PDR.

Variable	Percentage	95% CI
Age (years)		
15–24	33.31	32.23, 34.38
25–44	41.04	39.94, 42.15
45–64	19.21	18.42, 19.98
65+	6.44	5.91, 6.97
Gender		
Male	49.21	48.09, 50.33
Female	50.79	49.67, 51.90
Residence		
Urban	31.75	31.12, 32.38
Rural	68.25	67.62, 68.87
Household Income		
<1 USD/day	12.05	11.30, 12.81
1–2 USD/day	16.77	15.98, 17.56
2–3 USD/day	19.11	18.28, 19.94
>3 USD/day	52.07	51.11, 53.01
Education		
None	15.00	14.20, 15.78
Preschool/Primary	36.89	35.86, 37.92
Lower/Upper Secondary	37.99	36.96, 39.03
Vocational/Middle School	3.92	3.52, 4.31
University/Post Graduate	6.20	5.72, 6.70

**Table 2 ijerph-16-03059-t002:** Associations between demographic variables and self-reported lifetime history of tuberculosis in 9043 adults (ages 15 and older) of the 2012 National Adult Tobacco Survey of Lao PDR.

Variable	Odds Ratio	95% CI
Gender		
Male	1.00	Referent
Female	0.75	0.60, 0.92
Residence		
Urban	1.00	Referent
Rural	1.31	0.49, 3.49
Household Income		
<1 USD/day	1.00	Referent
1–2 USD/day	0.29	0.13, 0.65
2–3 USD/day	0.69	0.32, 1.51
>3 USD/day	0.55	0.26, 1.18
Education		
None	1.00	Referent
Preschool/Primary	0.82	0.45, 1.50
Lower/Upper Secondary	0.48	0.24, 0.96
Vocational/Middle School	0.69	0.21, 2.34
University/Post Graduate	0.99	0.27, 3.62

**Table 3 ijerph-16-03059-t003:** Associations between self-reported exposure to tobacco smoke and varied sources of household and non-household air pollution (i.e., smoked tobacco and air pollution) and lifetime history of tuberculosis in 9043 adults (ages 15 and older) of the 2012 National Adult Tobacco Survey of Lao PDR.

Variable	Odds Ratio	95% CI
Smoked Tobacco		
None	1.00	Referent
All Forms	1.73	1.00, 2.98
Hand Rolled Cigarettes	1.68	0.90, 3.11
Manufactured Cigarettes	1.14	0.75, 1.75
Water Pipe	0.42	0.17, 1.04
Tobacco Pipe	21.03	5.53, 79.88
Air Pollution		
Household Sources:		
ETS * Exposure at Home		
None	1.00	Referent
Daily	0.84	0.42, 1.45
Weekly	1.12	0.63, 1.99
Monthly	1.16	0.45, 3.00
Indoor Cooking Fires		
No	1.00	Referent
Yes	1.61	0.68, 3.81
Outdoor Cooking Fires **		
No	1.00	Referent
Yes	0.72	0.31, 1.66
Trash Burning		
No	1.00	Referent
Yes	1.69	0.92, 3.11
Non-household Sources (i.e., community, occupational):		
ETS Exposure in Enclosed Work Areas		
Less than Monthly	1.00	Referent
At least Monthly ***	1.93	0.85, 4.41
Crops Burning		
No	1.00	Referent
Yes	0.93	0.33, 2.63
Other Outdoor Fires		
No	1.00	Referent
Yes	2.30	1.15, 4.59

* Environmental tobacco smoke; ** Usually within 1–2 m of the house; *** Exposure within the last 30 days.

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
