# Peer review of "Smoked Tobacco, Air Pollution, and Tuberculosis in Lao PDR: Findings from a National Sample"

_ijerph, 2019, doi:10.3390/ijerph16173059_

Round 1

Reviewer 1 Report

The strongest points of this paper are the large sample size and the up to date references. However, the statistical analysis is disappointing as it is almost exclusively composed of descriptive methods. A much more robust, inference methods approach should be applied in order to turn the paper into a publishable one.

Author Response

We thank the reviewer the comments.  In response, we note that our aim with this paper was to specifically test the hypothesis that smoking and sources of indoor and outdoor air pollution significantly add to the burden of air pollution in Lao PDR.

Our sub-aims (tested through inferential methods of association):

1) To determine whether smoked tobacco contribute to higher rates of TB

2) To determine whether indoor air pollution sources (cooking fires, ets) contribute to higher rates of TB.

3) To determine whether outdoor air pollution sources (crop burning, trash fires) contribute to higher rates of TB.

4) To determine whether the combination of smoking, indoor air pollution, and outdoor air pollution incrementally contribute to higher rates of TB.

Our findings indicate significant contribution from 1) to 3) and that there is about a 47% increase in rates of TB per environmental exposure in the case of multiple exposures.

In the revised manuscript we have emphasized our aims, hypotheses, and highlighted how we have used inferential, multivariable methods (logistic regression tests of exposure and trend)  to test for an effect.

Reviewer 2 Report

This is an interesting article although the results are not surprising

The use of ACRONYMS is excessive and makes the reading of the article unnecessarily difficult for the reader who may not be familiar with the area 

It is not clear that the association between environmental tobacco smoke exposure and lifetime history of TB was present in people who had never smoked as well as in active smokers

The suggestion that the significant association between ETS exposure and TB history may result from a person living or working with a smoker who is at increased risk of transmitting the infection does not appear to have been considered    

Author Response

The use of ACRONYMS is excessive and makes the reading of the article unnecessarily difficult for the reader who may not be familiar with the area:

Point well taken. Have significantly reduced use of acronyms (i.e., IAP, LTBI, PSU, EA, LMIC, NIS, WPR), retaining the use of only three primarly acronyms (i.e., TB, ETS, and NATSL).

It is not clear that the association between environmental tobacco smoke exposure and lifetime history of TB was present in people who had never smoked as well as in active smokers.

This was tested in the combined analyses were subjects were given 1 point for smoking alone or ETS alone, and then additional points for the addition of the other exposure.  In the revised manuscript we have also tested it more directly with a run of ETS related to TB among never or non-smokers.

The suggestion that the significant association between ETS exposure and TB history may result from a person living or working with a smoker who is at increased risk of transmitting the infection does not appear to have been considered.

We mention this in the limitations section of the revised manuscript.

Reviewer 3 Report

This paper is a nice review of the topic as well as providing new data support for the influence of smoking on TB. The results for HAP were a bit surprising, but important to report.

Please can the HAP references used for 37, 38, and 39 be re-selected from the current literature. The HAP sector has changed dramatically since 2008 and there have been new demographic reports from IHME.

Author Response

Thank you for bring this to our attention. Reference # 37-39 have been replaced with more recent reports.

Reviewer 4 Report

Overall this is a well written manuscript and an area of research that has a significant impact on human health. The presentation of the tables could be improved and further interpretation of the data would strengthen the manuscript. Specific comments are:

Table 3: “inhaled particulate matter” is inaccurate as there are likely mainly volatile compounds and gases inhaled from these exposures as well

                 Why is there not a referent for the ETS Exposure at home?

Discussion on the use of binary measurements should also be included as a limitation of this analysis.

Overall the discussion is lacking interpretation of the results.

The authors make assumptions about their “measurement” of air pollution, this implies that actual quantification has occurred which does not appear to have been conducted. Phrasing throughout should be adjusted to clearly describe this as presence of … not measurement of….

Author Response

Table 3: “inhaled particulate matter” is inaccurate as there are likely mainly volatile compounds and gases inhaled from these exposures as well:

This phrase has been changed to: “tobacco smoke and varied sources of household and non-household air pollution” (Please see page 8; Table 3

Why is there not a referent for the ETS Exposure at home?

There is a referent category for this binary exposure and we have clarified that in the results section of the revised manuscript.

Discussion on the use of binary measurements should also be included as a limitation of this analysis.

We have added the limitations of the binary measures to the limitations section.

Overall the discussion is lacking interpretation of the results. The authors make assumptions about their “measurement” of air pollution, this implies that actual quantification has occurred which does not appear to have been conducted. Phrasing throughout should be adjusted to clearly describe this as presence of … not measurement of….:

We have changed the wording to aid interpretation of the strength of the results.

Round 2

Reviewer 1 Report

The overall quality of the article has improved. However there is still a methodological question that, in my view, must be addressed. The logistic regression has a number of requirements in order to be applied. The author should, therefore, include in the paper the validations of those data requirements in order to assure the reader that the logistic regression can be applied. Otherwise it is not the adequate method.

Author Response

We thank the reviewer for the important comments on methods.

The logistic regression is being applied to a sample of 9,043 subjects with 151 events (we reported the event rate of 1.67% for TB events in the current version of the manuscript).  Under assumptions of logistic regression (10 events per variable) this allows us to run models with up to 15 variables in the model.  Our analyses in the paper do not exceed that limit.  To test the fit of continuous variables in the model we used the Lemeshow Hosmer Statistic.  To test the fit of dummy variables we used the log likelihood ratio test.

We have added these points to the methods section of the manuscript to respond to the reviewer's comment.